# Diversity of bacterial community in the rhizosphere and bulk soil of *Artemisia annua* grown in highlands of Uganda

**Namuli Aidah**[1,2]*, **Ogwang Patrick Engeu**[1,2], **Tumuhairwe John Baptist**[3], **Vincent B. Muwanika**[4], **Mayega Johnson**[4], **Bazira Joel**[1,5]

**1** Pharmbiotechnology and Traditional Medicine Center of Excellence, Mbarara University of Science and Technology, Mbarara, Uganda, **2** Department of Pharmacy, Mbarara University of Science and Technology, Mbarara, Uganda, **3** Department of Agricultural Production, Makerere University, Kampala, Uganda, **4** Molecular Biology Laboratory Institute of Environment and Natural Resources, Makerere University, Kampala, Uganda, **5** Department of Microbiology, Mbarara University of Science and Technology, Mbarara, Uganda

* namuli_aidah@yahoo.com

**Data Availability Statement:** NCBI BioSample accession SAMN20588857.

**Funding:** The author(s) received no specific funding for this work.

## Abstract

High land areas in Uganda are suitable for the farming of *Artemisia annua*. However, harvested *A. annua* from these areas contain varying concentrations of antimalarial components. This may be attributed to variation in soil properties which affect vegetative growth characters, yield and active compounds of *A. annua*. Thus, bacterial composition and physiochemical properties of soil from Kabale and Kabarole high land areas where *A. annua* is grown were studied. The study objective was to determine the diversity of bacterial community in the rhizosphere and bulk soil of *A. annua* grown in highlands of Uganda. Composition of bacterial community was analyzed by amplicon sequencing of 16S rRNA genes on an Illumina Miseq platform. A total of 1,420,688 read counts was obtained and clustered into 163,493 Operational Taxonomic Units ((OTU). Kabarole highland had more OTUs (87,229) than Kabale (76,264). The phylum Proteobacteria (34.2%) was the most prevalent followed by Acidobacteria (17.3%) and Actinobacteria (15.5%). The bacteria community in the two highlands significantly differed (p <0.05) among all phyla except Proteobacteria. The main genera in bulk soil were povalibacter, brevitalea, nocardioides, stenotrophobacter, gaiella and solirubrobacter. Sphingomonas, ramlibacter paludibaculum and pseudarthrobacter were the main genera in *A. annua* rhizospheric soil.

## Introduction

*Artemisia annua* is widely grown in different parts of the world as a cheap source of the antimalarial compounds such as artemisinin, flavonoids, aromatic oils and polysaccharides [1, 2]. In Uganda, *A. annua* was introduced around 2003 [3] and is mainly grown in Wakiso, Kaberamaido, Kapchorwa, Rukungiri, Kabarole and Kabale districts. Itis cultivated as a monocrop or intercropped with beans. The main districts where the crop is grown include. The content of

**Competing interests:** The authors have declared that no competing interests exist.

active compounds varies greatly depending on the geographical location. The highland areas produce *A. annua* with more active compound than low land areas [3, 4]. For instance, they [3] observed that artemisinin and total flavonoids levels were higher in samples obtained from high land areas (Kabarole and Kabale) compared to those obtained from lowland regions (wakiso) i.e. 0.8%, 0.5% Vs 0.4% and 2.6%, 2.55% Vs 1.5% respectively. However, the artemisinin concentration is very low compared to other parts of the world that produce *A. annua* with upto 2% artemisinin [5]. Improving the concentrations of the antimalarial compounds in *A. annua* grown in Uganda is therefore an important area of investigation. Various plant growth promoting bacteria (PGPB) such as *Azotobacter*, *Azospirillum*, *Bacillus* and *Pseudomonas* have been reported to increase the concentration of artemisinin [6, 7] elsewhere. However, there is no study that highlights the rhizobacterial community of *A. annua* yet understanding the rhizobacterial community of a given plant species is a vital when considering the use of rhizobacteria as plant growth promoters [8]. This study therefore aimed at profiling the diversity of bacterial community in the rhizosphere and bulk soil of *A. annua* grown in highlands of Uganda as basis for the use of microbial inoculants to enhance its antimalarial compounds.

## Methodology

### Study sites and sample collection

The study was conducted in 2 highlands of Uganda i.e. Kabale (South Western Uganda) and Kabarole (Western Uganda). These are areas producing large volumes of artemisinin ranging from 0.5 and 0.8% [3]. The altitude of Kabarole and Kabale district is 1300–3800 meters and 2,000 meters above sea level respectively. Kabarole and Kabale district receive annual rainfall ranging from 1,200 mm– 1,500 mm and 800 mm– 1000 mm respectively [9, 10]. Soil samples were collected at the time of harvesting *A. annua*. Rhizosphere and bulk soils were sampled from the four existing cropping types of (i) intercrop of *A. annua* and beans (AA+B), (ii) beans alone (B), (iii) *A. annua* alone (A) and iv) Control-No crop grown, (N AA/B)). For the physicochemical analysis, composite samples (each 1.5 kg) were picked from the top soil (0–15 cm). For each of the 4 cropping type, 4 farms in each of the 2 districts having similar treatments were considered as replicates, thus bringing the total to 32 samples. The composite sample consisted of 10 cores obtained using zigzag technique. All 32 samples were placed in clean labelled sealable plastic bags and transported to the laboratory with in a day of collection. The soil was dried and sieved using a 2 mm sieve.

For DNA extraction and molecular analyses, samples were put in collecting tubes (2.2g each) and were stored at −80˚C. For each of the 4 cropping types, 4 farms in each of the 2 districts having similar treatments were considered as replicates and in each replicate 4 plants were selected to obtain rhizospheric sample (where *A. annua* plants were grown) or 4 cores were selected for obtaining bulk samples (where there was no *A. annua*).

### Laboratory analysis

**Physicochemical analysis.** Various soil properties such as pH, texture, organic matter, N, P and K were analyzed. Soil pH was measured with a pH meter in water (1:10 w/v). Soil texture was determined using Bouyoucos method [11, 12]. The organic matter was determined using Walkley and Black method [13]. The total nitrogen content was determined using Kjeldahl method [14] after total digestion of soil sample. The soluble phosphorus was determined using Ascorbic acid method [15]. Exchangeable potassium was extracted in ammonium acetate and measured using a flame photometer [16].

**DNA extraction.** Genomic DNA (gDNA) was isolated from each soil sample (0.25g) using Qiagen DNeasy ®Power soil ® kit (Germany). After each extraction, agarose electrophoresis

**Table 1. Summary of results of preprocessing and clustering.**

| Sample Count | Read Count | Gamma-diversity | Counts/sample | | | |
|---|---|---|---|---|---|---|
| | | | Min | Max | Median | Mean |
| 128 | 1,420,688 | 11,125 | 4,142.0 | 15,791.0 | 11,090.5 | 11,099.1 |

| Filtered Read Count | | | | | | |
|---|---|---|---|---|---|---|
| Ambiguous | Wrong prefix or primers | Sequence of prefix or primer (V-Region) | Low quality | Chimera | Other | |
| 144,960 | 869,637 | CCTACGGG[ACGT]GGC[AT]GCAG (V3-V4) | 127,642 | 435,721 | 9,351,007 | |

was conducted to detect the presence of gDNA. Later, the concentration of gDNA was quantified using Qubit 3 Fluorometer (Singapore) and an average of 6.8 ng/µl was obtained. Thereafter, 14 µls of gDNA was sent to Macrogen for amplicon based metagenomics sequencing.

*Assembly*. Longer assembled reads were obtained using the FLASH (1.2.11) program [17]. On average for each sample, the results of assembly were 43,859,440 (total bases), 96,482 (read count), 0.00 (N, %), 56.61(GC, %), 97.43 (Q20, %) and 90.43 (Q30, %).

*Preprocessing and clustering*. This was carried out using two programs i.e. CD-HIT-OTU MiSeq/FLX [18] and rDNA tools-PACBIO [19]. The steps involved identifying of chimeric reads and removing them, filtering out of short reads, trimming of extra-long tails. Filtered reads were clustered at 100% identity using CD-HIT-DUP. Secondary clusters were recruited into primary clusters. Noise sequences in clusters of size X or below were removed and X was statistically calculated. Remaining representative reads from non-chimeric clusters were clustered using a greedy algorithm into OTUs at a user-specified OTU cut off (e.g. 97% ID at species level). The results of preprocessing are summarized in Table 1.

*Taxonomic assignment and diversity statistics*. This was carried out using the program QIIME [20]. A reasonable number of reads was used in analysis since the rarefaction curves for the various samples became flatter to the right. Representative sequences from each OTUs were used to assign taxonomy. Furthermore, to identify differences between various treatments and the two study sites, data was analyzed statistically by one-way analysis of variance and independent student t- test ($p < 0.05$) using SPSS 21.0 Software (SPSS, Chicago, IL, USA). In addition, Principal Component Analysis (PCA) was conducted to find the lowest number of factors which could account for the variability in the original variables that were associated with those factors.

*Genera abundance*. A total of 626 bacterial genera (abundance $\geq 0.01\%$) were observed among the 3 most prevalent phylum (proteobacteria, acidobacteria and actinobacteria) common to both districts. However, PCA was conducted on 31 genera species that showed higher abundance ($\geq 0.2\%$) and the physiochemical properties of soil. The suitability of data for factor analysis was assessed by running the correlation matrix which revealed the presence of many coefficients of 0.3 and above. Furthermore, since variables were of various units (%, mg/l), they were first normalized [21] in order to bring the values of the different variables within the comparable range. This was done by subtracting the mean from the observed value and dividing by the standard deviation for each 42 variables using the following formula.

$$\text{Normalized value} = \frac{\text{Observed value} - \text{Mean}}{\text{Standard deviation}}$$

[21]

Having standardized the data, weights were attached using Principal Component Analysis (PCA) for all the soil properties in STATA 15 Statistical Software for assigning the weights. The loadings from the first component of PCA are used as the weights for the indicators. The assigned weights varied between -1 and +1, sign with the magnitude of the weights describing the contribution of each variable to the overall value of the index for soil properties. Two statistical tests were first conducted in order to determine the suitability of PCA. First, the Kaisers-Meyer-Olkin (KMO) measure of sampling adequacy score whose value of 0.811 was above the minimum recommended level of 0.60 [22] and Bartlett's Test of sphericity [23] was 0.000<0.01 implying that it was statistically significant thus supporting the factorability of the correlation matrix. Determination of the number of components to retain was based on Eigen value greater than 1.0 and the scree plot. Principal component Analysis revealed the presence of nine components with eigenvalues exceeding 1 and explained 88.32% as a cumulative of the variance. However, five components were first used in analysis and the components that contributed smaller amounts of the total variance were left out. Then finally, using Catell's Scree plot [24], it was decided to retain two components for further investigation. The two-component solution explained a total of 56.34% of the variance, with Component 1 contributing most variance (47.61%) and Component 2 explaining the least (8.73%). To aid in the interpretation of these two components, varimax rotation was performed. Using information about how much of the variance is explained by each item (the communalities table), soil components that had small values less than 0.3 and thus failed to load on the components obtained, an indication that they did not fit well with other items in its component were discarded. Furthermore, a Rotated Factor Matrix table was constructed to tell us what the factor loadings looked like after rotation.

## Results

### Physiochemical properties

Results obtained from analysis of physical and chemical properties of soil are presented in Table 2. The soils in Kabale are clay loam and have neutral pH (6.64) while Kabarole soil are sandy loam and have slightly alkaline pH (7.48). The content of Nitrogen, Potassium, Sodium

**Table 2. Physical and chemical properties of soil in the study sites.**

| KABALE | | | | | | | | | |
|---|---|---|---|---|---|---|---|---|---|
| Treatments | pH | OC | N | P | K | Na | Mg | Ca | Textural class |
| B | 6.50±0.44[a] | 2.33±1.20 [a] | 0.24±.12 [a] | 4.09±1.46 [b] | 0.49±0.29 [a] | 0.44±0.64 [a] | 5.07±2.65 [a] | 8.07±9.58 [a] | Clay loam |
| NA/B | 6.50 ±0.83 [a] | 2.87±1.16 [a] | 0.34±.16 [a] | 10.43±13.29 [a] | 0.63±0.65 [a] | 0.14±.06 [a] | 4.79±2.62 [a] | 14.83±13.06 [a] | Clay loam |
| A+B | 6.93±0.69 [a] | 2.85±1.27 [a] | 0.34±.19 [a] | 10.71±15.10 [a] | 0.56±0.31 [a] | 0.73±1.18 [a] | 6.84±2.93 [a] | 26.32±14.78 [a] | Clay loam |
| A | 6.62±0.77 [a] | 3.13±2.31 [a] | 0.35±.20 [a] | 14.64±17.23 [a] | 0.93±0.76 [a] | 0.42±0.60 [a] | 10.94±10.92 [a] | 31.36±41.74 [a] | Clay |
| KABAROLE | | | | | | | | | |
| B | 7.97±0.30 [a] | 3.88±0.70 [a] | 0.37±0.07 [a] | 475.10±74.64 [a] | 0.88±0.83 [a] | 0.35±0.00 [a] | 14.51±3.24 [a] | 115.43±6.95[a] | Loam |
| NA/B | 7.46±0.11 [a] | 3.70±0.48 [a] | 0.42±0.07 [a] | 410.61±228.92 [a] | 0.70±0.50 [a] | 0.32±0.13 [a] | 12.30±3.40 [a] | 84.07±19.46 [b] | Sandy Loam |
| A+B | 7.27±0.62 [a] | 4.09±0.89 [a] | 0.43±0.11 [a] | 669.01±377.79 [a] | 0.96±0.78 [a] | 0.33±0.09 [a] | 12.95±5.26 [a] | 85.77±37.97 [b] | Sandy Loam |
| A | 7.22±0.80 [a] | 3.46±0.24 [a] | 0.38±0.03 [a] | 310.87±172.05 [a] | 0.41±0.30 [a] | 0.25±0.04 [a] | 6.22±1.54 [a] | 54.96±28.47 [c] | Sandy Loam |

Values are expressed as mean ± standard deviation, for each study site, mean values in the same column having different letters are significantly different using ANOVA (p< 0.05) and multiple comparison of means was done using Least Significant Difference (LSD) (n = 16). B-beans only, NA/B- no beans and no *A. annua*, B+A- Beans and *A. annua* and *A. annua* only.

and Silt is similar in both Kabale and Kabarole. However, the content of Phosphorus, Organic Carbon, Magnesium and Calcium were higher in Kabarole soils than Kabale soils. Bulk soils had lower physiochemical properties than rhizospheric soils in Kabale.

## Phyla abundance

The phylum Proteobacteria was the most prevalent followed by Acidobacteria and Actinobacteria (Fig 1A and 1B). Comparing the two districts, Kabale soils had higher diversity of bacteria than Kabarole and there was significant difference (P≤0.5) among all phyla except phylum Proteobacteria. Kabale soil samples contained more Acidobacteria, Bacteroidetes, Veruccomicrobia, Gemmatimonadetes and Firmicutes while Kabarole soil samples contained more Actinobacteria and other phyla like Plactomycetes. With respect to the various soil treatments, there was no significant difference (P≤ 0.05) among the abundance of various phyla in Kabale. However, in Kabarole, there was significant differences (P≤ 0.05) in the abundance of phyla actinobacteria (14.85%) and verrucomicrobia (7.01%) in soils with artemisia only and the other treatments. Comparing bulk and rhizospheric soils, Proteobacteria were dorminant in both soils but acidobacteria and actinobacteria were more dorminate in bulk soils of Kabale and Kabarole respectively.

**Genus abundance.** The abundance of various bacterial genera (≥ 0.2%) in the bulk and rhizospheric soil of the most prevalent phylum (Proteobacteria, acidobacteria and actinobacteria) are shown in Table 3. Most genera in Kabale soils did not show any significant differences. However, most genera in Kabarole soils showed significant differences. In bulk soil, the genera that showed significant differences were po, br, no, str, ga and sol. In rhizospheric soils, Many genera (sps, ac, ram, ma, rhi, aci, ed, si, oc, pa and pse) showed significant differences with the bulk soil but genera sps, ram, pa and pse were observed to show significant in differences in both Kabale and Kabarole soils. Results of PCA of various soil components (genera and physiochemical properties) are shown in Fig 2. Seven genera (sps, Lu, he, st, ma, spm and pse) had positively higher loading with the second component. Twenty nine soil components had higher loading with the first component and 18 (sand, Ca, Mg, P, N, OCa, pH, Sol, ga, Str, Vi, acis, pe, az, ram, ral, P and Ly) had positive factor loading while 11 (ac, ni, ps, aci, rhi, ed, si, oc, pa, pa_a and clay) had negative factor loading.

## Discussion

The most abundant phylum of the *A. annua* rhizospheric bacterial community were Gemmatimonadetes, Acidobacteria and Proteobacteria. With the exception of acidobacteria, the results varied with what was reported by [25] as they observed Chloroflexi, Cyanobacteria, and Planctomycetes as the most abundant in *A. annua* rhizosphere soil yet in this study, they were the least abundant. The variation may be stemming from the fact that plants were of different varieties and were also grown in different soil types at different altitudes.

Proteobacteria have been reported to be dorminant in nutrient rich soil (copiotrophic) while acidobacteria are dominant in oligotrophic conditions [26] and in soils with lower pH [27]. Thus the results obtained tallied with what has been reported. Both bulk and rhizospheric soils had copiotrophic conditions as the ratio of proteobacteria to acidobacteria was high [26] and also the nutrients were high (Table 2). Furthermore, bulk soils of Kabale had lower physiochemical properties and thus were expected to have more acidobacteria than rhizospheric soils and this was what was observed. On the other hand, more actinobacteria was observed in slightly alkaline Kabarole sandy loam soils that were rich in organic matter than in neutral Kabale soils, this observation tallied with reports of [28, 29].

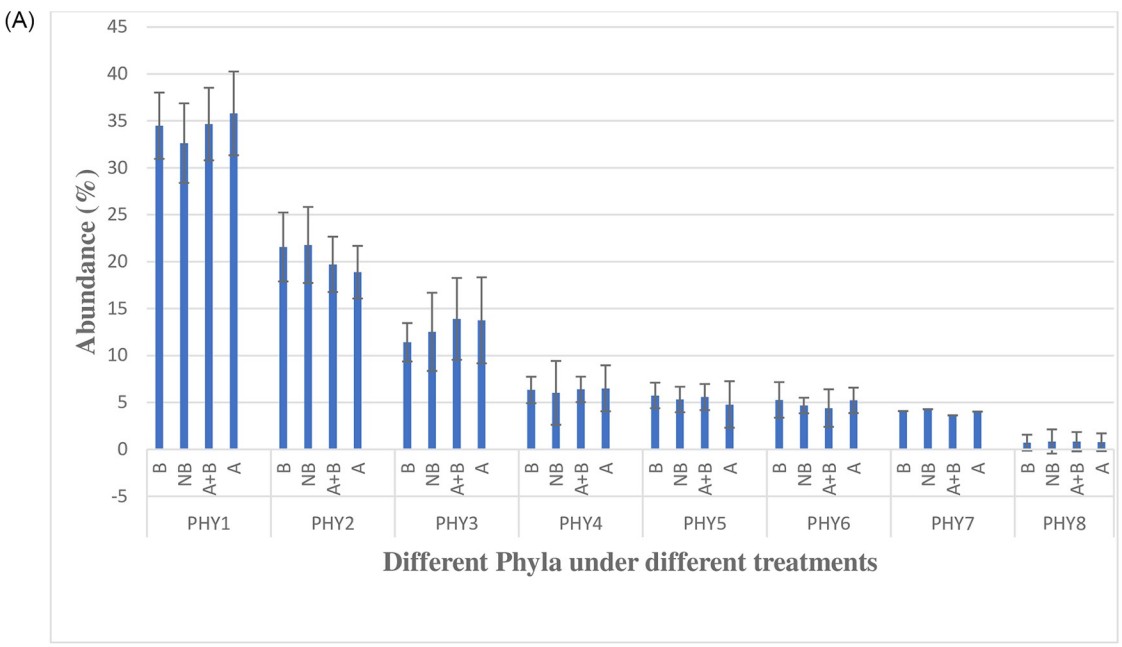

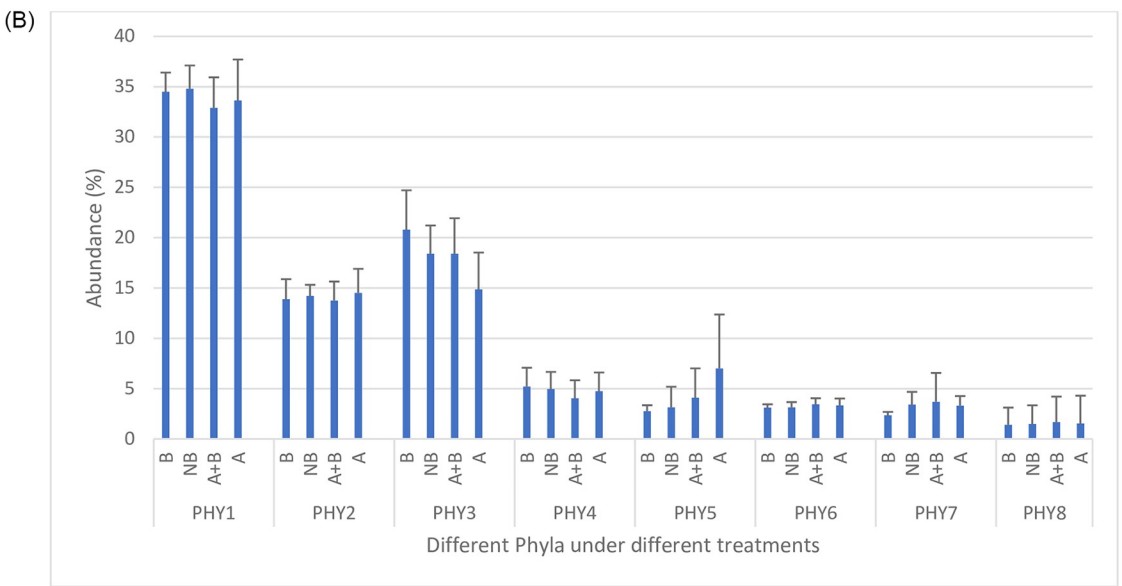

**Fig 1.** **A:** Proportion of major bacterial phyla found in the four soil treatment samples obtained from Kabale district. B- Beans Only, NBA- No Beans and *Artemisia*, N+B- Beans and *Artemisia* and A-*Artemisia* Only (B and NBA-bulk soil, A+B and A- rhizospheric soil). PHY1- Proteobacteria, PHY2- Acidobacteria, PHY3- Actinobacteria, PHY4- Bacteroidetes, PHY5- Verrucomicrobia, PHY6-Gemmatimonadetes, PHY7- Firmicutes, PHY8- Others. **B:** Proportion of major bacterial phyla found in the four soil treatment samples obtained from Kabarole district. B- Beans Only, NBA- No Beans and *Artemisia*, N+B- Beans and *Artemisia* and A-*Artemisia* Only (B and NBA-bulk soil, A+B and A- rhizospheric soil). PHY1- Proteobacteria, PHY2- Acidobacteria, PHY3-Actinobacteria, PHY4- Bacteroidetes, PHY5- Verrucomicrobia, PHY6-Gemmatimonadetes, PHY7- Firmicutes, PHY8- Others.

According to PCA, the genera observed were categorized basing on pH and PCA1 was alkaline pH and PCA2 was neutral pH. This so because genus Pse (had the highest positive loading to PCA2) has been reported to grow at an optimum pH of 7.0–7.5 [30]. Genera Rhi, aci and ed (had the highest negative loading to PCA1, greater than 0.9) have been reported to survive in

**Table 3. Percentage abundance ($\geq$ 0.2%) of various bacterial genera detected in bulk and rhizospheric soil.**

| Genus | Kabale[P] | | | | Kabarole[Q] | | | |
|---|---|---|---|---|---|---|---|---|
| | Bulk Soil | | Rhizospheric Soil | | Bulk Soil | | Rhizospheric Soil | |
| | B | NB/A | B+A | A | B | NB/A | B+A | A |
| Sps | 4.29 ± 1.28[ab] | 3.63 ± 1.17 [b] | 4.72 ± 2.48 [ab] | 5.56 ± 2.87 [a] | 2.97 ± 0.71 [ab] | 2.89 ± 1.01[b] | 2.86 ± 1.11[b] | 3.76 ± 1.73 [a] |
| Ac | 0.49 ± 0.55 [ab] | 0.81 ± 1.09 [a] | 0.27 ± 0.22 [b] | 0.29 ± 0.29 [b] | 0.10 ± 0.06[b] | 0.15 ± 0.06 [ab] | 0.17 ± 0.10 [ab] | 0.22 ± 0.19 [a] |
| Lu | 0.05 ± 0.04 [b] | 0.11 ± 0.19 [a] | 0.04 ± 0.03 [b] | 0.30 ± 0.59 [a] | 0.06 ± 0.05 [a] | 0.06 ± 0.06 [a] | 0.04 ± 0.04 [a] | 0.05 ± 0.02 [a] |
| Ly | 0.62 ± 0.38 [a] | 0.77 ± 0.41 [a] | 0.49 ± 0.33 [b] | 0.72 ± 0.33 [a] | 1.18 ± 0.18 [a] | 1.16 ± 0.70 [a] | 0.83 ± 0.34[b] | 0.88 ± 0.33 [ab] |
| ps | 0.46 ± 0.37 [a] | 0.54 ± 0.36 [a] | 0.36 ± 0.19 [a] | 0.59 ± 0.80 [a] | 0.32 ± 0.15 [ab] | 0.18 ± 0.09[b] | 0.28 ± 0.21 [ab] | 0.41 ± 0.31[a] |
| po | 0.70 ± 0.22 [a] | 0.72 ± 0.43 [a] | 0.71 ± 0.26 [a] | 0.76 ± 0.50 [a] | 1.69 ± 0.47 [a] | 1.51 ± 0.39 [ab] | 1.12 ± 0.79[b] | 1.09 ± 0.96[b] |
| ral | 0.13 ± 0.09 [a] | 0.24 ± 0.20 [a] | 0.23 ± 0.23 [a] | 0.25 ± 0.33 [a] | 1.38 ± 0.69 [a] | 1.26 ± 0.77 [ab] | 0.77 ± 0.56[b] | 1.00 ± 1.13 [ab] |
| ram | 0.59 ± 0.26 [ab] | 0.37 ± 0.20 [b] | 0.65 ± 0.25[a] | 0.75 ± 0.49 [a] | 0.98 ± 0.40[bc] | 0.94 ± 0.21[c] | 1.45 ± 1.14[ab] | 1.75 ± 0.60 [a] |
| he | 1.86 ± 0.48 [ab] | 1.33 ± 0.41 [c] | 2.22 ± 0.69 [a] | 1.78 ± 0.61 [b] | 1.08 ± 0.28 [a] | 0.96 ± 0.29 [a] | 0.95 ± 0.58 [a] | 0.93 ± 0.19 [a] |
| ma | 0.90 ± 0.45 [a] | 0.52 ± 0.34 [c] | 0.80 ± 0.40[ab] | 0.63 ± 0.36[bc] | 0.34 ± 0.14[b] | 0.37 ± 0.15[b] | 0.83 ± 1.05 [ab] | 1.23 ± 1.00 [a] |
| rhi | 1.30 ± 0.93[ab] | 1.78 ± 1.65 [a] | 0.76 ± 0.48 [b] | 1.05 ± 1.13 [ab] | 0.17 ± 0.06[b] | 0.19 ± 0.07[b] | 0.41 ± 0.51 [ab] | 0.60 ± 0.55 [a] |
| rho | 1.76 ± 0.47 [a] | 1.83 ± 0.43 [a] | 1.97 ± 0.64 [a] | 1.88 ± 0.63 [a] | 1.57 ± 0.26[b] | 1.89 ± 0.58 [a] | 1.79 ± 0.32 [ab] | 1.66 ± 0.43 [ab] |
| pe | 0.17 ± 0.13 [a] | 0.22 ± 0.17 [a] | 0.27 ± 0.17 [a] | 0.25 ± 0.15 [a] | 1.06 ± 0.33[b] | 1.34 ± 0.15 [a] | 1.19 ± 0.54 [ab] | 0.73 ± 0.24[c] |
| spm | 0.16 ± 0.49 [a] | 0.07 ± 0.05 [a] | 0.18 ± 0.11 [a] | 0.20 ± 0.10 [a] | 0.08 ± 0.06 [a] | 0.08 ± 0.04 [a] | 0.17 ± 0.26 [a] | 0.14 ± 0.16 [a] |
| aci | 4.38 ±3.04 [ab] | 5.57 ± 5.25 [a] | 2.53 ± 1.92 [b] | 2.84 ± 2.56 [b] | 0.21 ± 0.11 [b] | 0.29 ± 0.18[b] | 0.97 ± 1.28 [ab] | 1.70 ± 1.77 [a] |
| ed | 1.52 ± 1.59 [a] | 1.24 ± 1.16 [abc] | 0.70 ± 0.92 [c] | 0.63 ± 0.62 [bc] | 0.13 ± 0.06[b] | 0.13 ± 0.04[b] | 0.20 ± 0.11 [ab] | 0.29 ± 0.19 [a] |
| Si | 0.22 ± 0.38 [bc] | 0.64 ± 0.83 [a] | 0.09 ± 0.16 [c] | 0.16 ± 0.20 [bc] | 0.00 ± 0.00[b] | 0.00 ± 0.00[b] | 0.01 ± 0.01 [ab] | 0.04 ± 0.06 [a] |
| oc | 0.54 ± 0.68[a] | 0.35 ± 0.37[ab] | 0.30 ± 0.54[ab] | 0.15 ± 0.25[b] | 0.00 ± 0.01[b] | 0.00 ± 0.01[b] | 0.04 ± 0.09 [ab] | 0.07 ± 0.09 [a] |
| pa | 4.56 ± 2.16 [a] | 3.91 ± 1.71 [a] | 4.06 ± 2.17 [a] | 3.51 ± 1.31 [a] | 0.90 ± 0.22[b] | 0.99 ± 0.33[b] | 1.21 ± 0.51 [ab] | 1.53 ± 0.61 [a] |
| st | 0.79 ± 0.23 [a] | 0.75 ± 0.47 [a] | 1.01 ± 0.56 [a] | 0.88 ± 0.45 [a] | 0.86 ± 0.29 [a] | 0.86 ± 0.22 [a] | 0.86 ± 0.28 [a] | 0.89 ± 0.36 [a] |
| br | 2.84 ± 2.81 [a] | 2.98 ± 2.52 [a] | 3.22 ± 1.97 [a] | 3.06 ± 1.65 [a] | 3.15 ± 0.43 [a] | 3.42 ± 0.59 [a] | 3.31 ± 0.83 [a] | 2.39 ± 0.73 [b] |
| vi | 4.78 ± 2.13 [a] | 4.82 ± 2.88 [a] | 5.97 ± 2.44 [a] | 6.01 ± 3.09 [a] | 6.97 ± 1.46 [a] | 7.10 ± 1.11 [a] | 5.85 ± 2.04 [a] | 6.09 ± 3.20 [a] |
| acis | 1.55 ± 0.32 [a] | 1.32 ± 0.21 [a] | 1.41 ± 0.34 [a] | 1.38 ± 0.46 [a] | 1.73 ± 0.26 [a] | 1.83 ± 0.23 [a] | 1.85 ± 0.24 [a] | 1.52 ± 0.27 [b] |
| pse | 0.82 ± 0.60 [bc] | 0.41 ± 0.28[c] | 1.77 ± 1.20 [a] | 1.41 ± 1.36[ab] | 0.81 ± 0.38[b] | 0.67 ± 0.19[c] | 0.95 ± 0.31 [ab] | 1.07 ± 0.50 [a] |
| no | 0.63 ± 0.30 [a] | 1.00 ± 1.07 [a] | 0.90 ± 0.47 [a] | 1.07 ± 0.58 [a] | 1.01 ± 0.40 [a] | 0.85 ± 0.23 [ab] | 0.73 ± 0.29[b] | 0.70 ± 0.42[b] |
| str | 0.75 ± 0.24 [a] | 0.93 ± 0.29 [a] | 0.95 ± 0.51 [a] | 0.83 ± 0.42 [a] | 2.34 ± 0.46 [a] | 2.04 ± 0.43 [ab] | 2.08 ± 0.53 [ab] | 1.41 ± 0.34[c] |
| ga | 2.12 ± 0.50 [a] | 2.47 ± 1.30 [a] | 2.36 ± 0.86 [a] | 2.46 ± 0.94 [a] | 6.88 ± 1.38 [a] | 5.62 ± 1.18 [b] | 5.69 ±1.51 [b] | 4.37 ±1.61 [c] |
| pa_a | 0.06 ± 0.06 [a] | 0.04 ± 0.06 [a] | 0.03 ± 0.03 [a] | 0.05 ± 0.06 [a] | 0.03 ± 0.02 [a] | 0.03 ± 0.02 [a] | 0.02 ± 0.03 [a] | 0.02 ± 0.01 [a] |
| ni | 1.56 ± 0.37 [a] | 1.53 ± 0.42 [a] | 1.53 ± 0.43 [a] | 1.66 ± 0.57 [a] | 1.06 ± 0.18 [a] | 0.91 ± 0.21[b] | 0.88 ± 0.25[b] | 1.12 ± 0.21 [a] |
| az | 0.30 ± 0.10[b] | 0.22 ± 0.18[b] | 0.58 ± 0.65 [a] | 0.27 ± 0.17[b] | 0.83 ± 0.27 [a] | 0.87 ± 0.36 [a] | 0.75 ± 0.52 [a] | 0.60 ± 0.50 [a] |
| sol | 0.58 ± 0.24[b] | 0.99 ± 0.64[a] | 1.00 ± 0.47 [a] | 0.76 ± 0.43[ab] | 1.83 ± 0.58 [a] | 1.58 ± 0.43 [ab] | 1.28 ± 0.43[bc] | 0.97 ± 0.48[c] |

Values are expressed as mean ± standard deviation, for each study site, mean values in the same row for each district having different letters are significantly different using ANOVA (p< 0.05) and multiple comparison of means was done using least significant difference (LSD) (n = 16). sps- Sphingomonas, ac- Acidibacter, lu- Luteimonas, ly- Lysobacter, ps-Pseudomonas, po- Povalibacter, ral- Ralstonia, ram- Ramlibacter, he- Herbaspirillum, ma- Massilia, ni- Nitrosospira, az- Azoarcus, rhi- Rhizomicrobium, rho- Rhodoplanes, pe-pedomicrobium, spm- Sphingobium, aci- Acidobacterium, ed- Edaphobacter, si- Silvibacterium, oc- Occallatibacter, pa- Paludibaculum, st- Stenotrophobacter, br- Brevitalea, vi- Vicinamibacter, acis- Aciditerrimonas, pse- Pseudarthrobacter, no- Nocardioides, str- Streptomyces, ga- Gaiella, pa_a- Patulibacter and sol- Solirubrobacter.

acidic pH not alkaline pH i.e. genus ed (4.0–7.0) [31], aci (4.5–7.0) [32] and rhi (5.5–7.3) [33]. On the other hand, also the genera with positive loading to PCA1 were not very high (had less than 0.9) as most of species of these genera have been observed to grow at moderately alkaline pH not at pH14.0 [34–36]. These results tallied with the findings of [37] as they indicated that soil pH is vital in determining which genera present in agricultural soils.

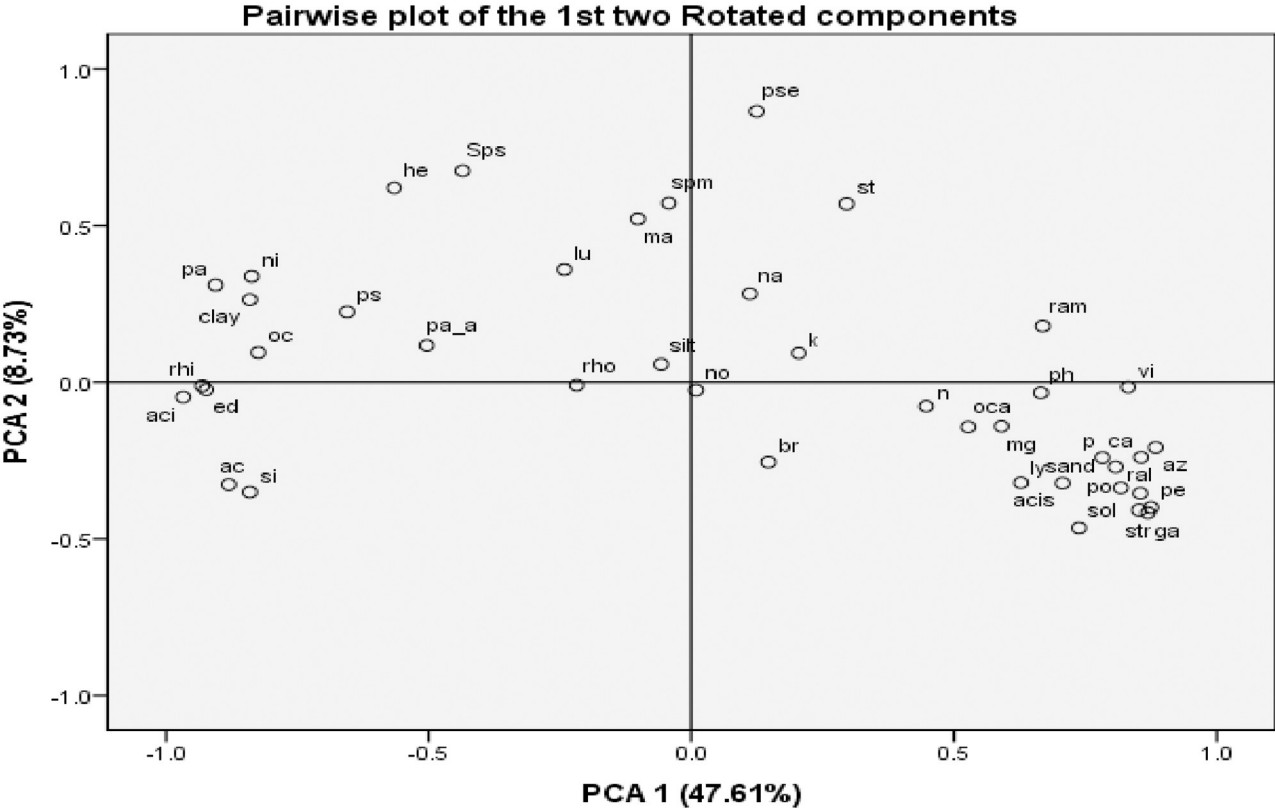

**Fig 2.** Plot of correlated variables by PCA1 and PCA2 (sps- Sphingomonas, ac- Acidibacter, lu- Luteimonas, ly- Lysobacter, ps-Pseudomonas, po- Povalibacter, ral- Ralstonia, ram- Ramlibacter, he- Herbaspirillum, ma- Massilia, ni- Nitrosospira, az- Azoarcus, rhi- Rhizomicrobium, rho- Rhodoplanes, pe-pedomicrobium, spm- Sphingobium, aci- Acidobacterium, ed- Edaphobacter, si- Silvibacterium, oc- Occallatibacter, pa- Paludibaculum, st- Stenotrophobacter, br- Brevitalea, vi- Vicinamibacter, acis- Aciditerrimonas, pse- Pseudarthrobacter, no- Nocardioides, str- Streptomyces, ga- Gaiella, pa_a- Patulibacter, sol- Solirubrobacter, ph-PH, na-Na (sodium), ca-Ca (calcium), mg-Mg (Magnesium), p-P(phosphorus).

## Conclusion

In conclusion, the results show that the *A. annua* rhizosphere is a large reservoir of bacteria that may be capable of many roles. Most of the species observed in the rhizosphere are not among the most frequently mentioned non symbiotic PGPB (*Azospirillum* sp., *Azotobacter* sp., *Bacillus* sp., *Pseudomonas* sp. etc) mentioned in various reports. However, many of the species belong to the phylum proteobacteria which constitutes most PGPB. Thus, use of selected bacteria especially proteobacteria may promote *A. annua* growth and increase its phytochemical contents.

## Acknowledgments

Authors appreciate Pharm-BioTechnology and Traditional Medicine Centre (PHARMBIO-TRAC- Mbarara University of Science and Technology), Associate Professor Masembe Charles (Makerere University) and Mr. Shahiid Kiyaga (Makerere University).

## Author Contributions

**Conceptualization:** Namuli Aidah, Ogwang Patrick Engeu, Tumuhairwe John Baptist, Bazira Joel.

**Data curation:** Namuli Aidah, Bazira Joel.

**Formal analysis:** Namuli Aidah, Tumuhairwe John Baptist, Mayega Johnson.

**Investigation:** Namuli Aidah, Vincent B. Muwanika, Mayega Johnson.

**Methodology:** Namuli Aidah, Tumuhairwe John Baptist, Vincent B. Muwanika, Mayega Johnson, Bazira Joel.

**Supervision:** Ogwang Patrick Engeu, Tumuhairwe John Baptist, Vincent B. Muwanika, Bazira Joel.

**Validation:** Namuli Aidah, Ogwang Patrick Engeu, Tumuhairwe John Baptist.

**Visualization:** Namuli Aidah, Bazira Joel.

**Writing – original draft:** Namuli Aidah, Mayega Johnson, Bazira Joel.

**Writing – review & editing:** Namuli Aidah, Ogwang Patrick Engeu, Tumuhairwe John Baptist, Vincent B. Muwanika, Mayega Johnson, Bazira Joel.

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
