## [Decision Letter · Decision Letter 0]

18 Nov 2021

PONE-D-21-28060Diversity of bacterial community in the rhizosphere and bulk soil of Artemisia annua grown in highlands of UgandaPLOS ONE

Dear Dr. namuli,

Thank you for submitting your manuscript to PLOS ONE. After careful consideration, we feel that it has merit but does not fully meet PLOS ONE’s publication criteria as it currently stands. Therefore, we invite you to submit a revised version of the manuscript that addresses the points raised during the review process. In particular, most criticisms are related to the quality of the text and the figures and some minor, but important, points regarding analysis.

We look forward to receiving your revised manuscript.

Kind regards,

Lorenzo Brusetti

Academic Editor

PLOS ONE

Journal Requirements:

Reviewers' comments:

Reviewer's Responses to Questions

**Comments to the Author**

1. Is the manuscript technically sound, and do the data support the conclusions?

Reviewer #1: Partly

2. Has the statistical analysis been performed appropriately and rigorously? 

Reviewer #1: Yes

3. Have the authors made all data underlying the findings in their manuscript fully available?

Reviewer #1: Yes

4. Is the manuscript presented in an intelligible fashion and written in standard English?

Reviewer #1: No

5. Review Comments to the Author

Reviewer #1: Review PONE-D- 21-28060

Diversity of bacterial community in the rhizosphere and bulk soil of Artemisia annua grown in highlands of Uganda

Title: the specie name should be in italics

Abstract: OTU is misspelled.

Introduction: the first 2 paragraphs are a bit repetitive and can be shorten in one. Sometimes the wording is heavy and the same concept could be expressed in a clearer way (see the last paragraph of the introduction).

Methods: Sequencing paragraph can be deleted. Analysis can also be omitted. Authors can start directly with assembly section. Processing and clustering OTU is misspelled. Taxonomic assignment Fig.1 does not show the rarefaction curves but the composition of the samples.

Results: Fig.1 need to be changed. Pie charts are easily misread, and the use of bar-chart is recommended. It would be easier to read if next to T1, T2 etc is added the description itself such as T1 Beans only or even T1 B as the authors used this code for table 2 and in the text. It is also not clear which test has been applied to test the effect of treatment across the different phyla composition, this should be clearly stated in the text and more info should be provided than the p-value alone such as R2, F value etc based on the test used by the authors. Same for table 2, authors clearly state the number of samples and used mean and standard deviation but not info of the type of test used were given (and data were tested for normality and variance?)

Species abundance:

Authors should reconsider their ability to assign OTUs to species levels. The clustering at 97% of identity is to determine the genus and not the species so authors should analysis the data at one taxonomic order higher (genera) and not species. All this section needs to be re-written based on the new data analysis generated on genus level.

Additional comments on this section: The description of how data were analysed for the PCA including KMO and Bartleets test should be moved to material and methods section. Here the authors should be presenting their results so for examples starting directly with the Principal compenten anyalis revelead etc... Again all the section talking about the varimax rotation should be moved to M&M section including the explanation why two species were removed (this part might not be any more present when data analysed at genus level). The section starting “Furthermore…” can be easily omitted and included in the M&M section.

Section relating Figure 2 and table 3, authors try to find a connection between the different species and the chemistry and physical properties of the soil based on the general composition of the samples, why did they not include these properties in the PCA analysis? This would support their statement better than the present statements.

Discussion:

First paragraph needs to be rewritten. Hard to follow and not providing any information more than a repetition of the results itself.

The paragraph about the species should be rewritten in the light of the use of genera.

Conclusion: no changes needed

General comments:

Attention OTUs misspelled all over the manuscript.

Phylum/phyla used in the wrong places.

Name of the phyla sometimes with capital letter sometimes with lower letter.

In general, the manuscript needs a revision to achieve a better fluidity in the text as many parts are disconnected and sometimes repetitive especially in the introduction.

6. PLOS authors have the option to publish the peer review history of their article (what does this mean?). If published, this will include your full peer review and any attached files.

Reviewer #1: No

---

## [Author Response · Author response to Decision Letter 0]

5 Apr 2022

Reviewer’s Comment Response

Title:

-The species name should be in italics 

-It was italicized

Abstract:

-OTU is misspelled.

-Changing species to genus in the last section of the result affected this section as the reviewer stated that “The clustering at 97% of identity is to determine the genus and not the species so authors should analysis the data at one taxonomic order higher (genera) and not species.” 

-OUT changed to OTU

-Different species replaced with different genera

Introduction:

 The first 2 paragraphs are a bit repetitive and can be shorten in one. Sometimes the wording is heavy and the same concept could be expressed in a clearer way (see the last paragraph of the introduction). 

First two paragraphs omitted and only the beginning part of second paragraph embedded at the beginning of the last paragraph

Methods: 

-Sequencing paragraph can be deleted. 

-Analysis can also be omitted. 

-Authors can start directly with assembly section. Processing and clustering OTU is misspelled. 

-Taxonomic assignment Fig.1 does not show the rarefaction curves but the composition of the samples.

-Section of writing from result to be moved to this section (species abundance) 

-Sequencing paragraph deleted

-Analysis paragraph omitted

-Began directly with assembly

- OUT changed to OTU in 2 positions i.e CD-HIT-OTU and a user-specified OTU

-Figure 1 deleted.

-Section of species abundance (changed to genus abundance) moved to this section.

Results:

Fig.1

Need to be changed. Pie charts are easily misread, and the use of bar-chart is recommended. It would be easier to read if next to T1, T2 etc is added the description itself such as T1 Beans only or even T1 B as the authors used this code for table 2 and in the text.

 It is also not clear which test has been applied to test the effect of treatment across the different phyla composition, this should be clearly stated in the text and more info should be provided than the p-value alone such as R2, F value etc based on the test used by the authors. 

Same for table 2, authors clearly state the number of samples and used mean and standard deviation but not info of the type of test used were given (and data were tested for normality and variance?)

Species abundance:

Authors should reconsider their ability to assign OTUs to species levels. The clustering at 97% of identity is to determine the genus and not the species so authors should analysis the data at one taxonomic order higher (genera) and not species. All this section needs to be re-written based on the new data analysis generated on genus level.

Additional comments on this section: The description of how data were analysed for the PCA including KMO and Bartleets test should be moved to material and methods section. Here the authors should be presenting their results so for examples starting directly with the Principal compenten anyalis revelead etc... Again all the section talking about the varimax rotation should be moved to M&M section including the explanation why two species were removed (this part might not be any more present when data analysed at genus level). The section starting “Furthermore…” can be easily omitted and included in the M&M section.

Section relating Figure 2 and table 3, authors try to find a connection between the different species and the chemistry and physical properties of the soil based on the general composition of the samples, why did they not include these properties in the PCA analysis? This would support their statement better than the present statements. 

---bar chart used instead 

-T1/T2—Disregarded and the initials used in table 2 were used

The test used was one way ANOVA, multiple comparison of means was done using LSD test.

(Was included in the methods and below table 1)

-R2 was not stated since the analysis used was ANOVA and not Ordinary least squares (OLS) regression

 -F-test is also under regression as it indicates whether the linear regression model provides a better fit to the data than a model that does not contain independent variables.

(In this paper, the interest was to determine if significant mean variations existed)

-Furthermore, one of the most essential requirements for use of ANOVA is the normality of the dependent variable. This assumption was checked and verified using histograms for all the variables. Generally, all the variables fulfilled the normality assumption because they had a normal distribution. As for the homogeneity of variance the Levene’s test was employed, all of which had insignificant values (p˃0.05) for Levene’s test i.e none of the variables violated the normality and homogeneity of variance tests.

Genus abundance:

-Reconsidered to genus level

-All this section was re-written based on the new data analysis generated on genus level.

Discussion:

First paragraph needs to be rewritten. Hard to follow and not providing any information more than a repetition of the results itself.

The paragraph about the species should be rewritten in the light of the use of genera.

References

-Paragraph one deleted

-paragraphs relating to species rewriiten in relation to genus

-Some few references were added as they were used to discuss the observations related to genera instead of species.

-Some few references were deleted as they were used to as references for information that was deleted (in the introduction and discussion)

---

## [Editor Report · Decision Letter 1]

26 May 2022

Diversity of bacterial community in the rhizosphere and bulk soil of Artemisia annua grown in highlands of Uganda

PONE-D-21-28060R1

Dear Dr. namuli,

We’re pleased to inform you that your manuscript has been judged scientifically suitable for publication and will be formally accepted for publication once it meets all outstanding technical requirements.

Kind regards,

Lorenzo Brusetti

Academic Editor

PLOS ONE
---

## [Editor Report · Acceptance letter]

24 Aug 2022

PONE-D-21-28060R1 

Diversity of bacterial community in the rhizosphere and bulk soil of *Artemisia annua* grown in highlands of Uganda 

Dear Dr. Aidah:

I'm pleased to inform you that your manuscript has been deemed suitable for publication in PLOS ONE. Congratulations! Your manuscript is now with our production department. 

Kind regards, 

on behalf of

Dr. Lorenzo Brusetti 

Academic Editor

PLOS ONE